# The Intrinsic Parameters of the Polyamide Nanofilm in Thin-Film Composite Reverse Osmosis (TFC-RO) Membranes: The Impact of Monomer Concentration

**DOI:** 10.3390/membranes12040417

**Published:** 2022-04-11

**Authors:** Mengling Zhang, Xiangyang Hu, Lei Peng, Shilin Zhou, Yong Zhou, Shijie Xie, Xiaoxiao Song, Congjie Gao

**Affiliations:** 1Center for Membrane Separation and Water Science & Technology, Ocean College, Zhejiang University of Technology, Hangzhou 310014, China; zhangmengling07@outlook.com (M.Z.); sun17816119566@163.com (X.H.); pl17858642890@163.com (L.P.); zsl18237065681@163.com (S.Z.); gaocj@zjut.edu.cn (C.G.); 2Bruker Shanghai Office 9F, Building NO.1, Lane 2570 Hechuan Rd, Minhang District, Shanghai 200233, China

**Keywords:** polyamide, intrinsic thickness, defects, permeability

## Abstract

The realistic resistance zone of water and salt molecules to transport across a TFC-RO membrane is the topmost polyamide nanofilm. The existence of hollow voids in the fully aromatic polyamide (PA) film gives its surface ridge-and-valley morphologies, which confuses the comprehensions of the definition of the PA thickness. The hollow voids, however, neither participate in salt–water separation nor hinder water penetrating. In this paper, the influence of intrinsic thickness (single wall thickness) of the PA layer on water permeability was studied by adjusting the concentration of reacting monomers. It confirms that the true permeation resistance of water molecules originates from the intrinsic thickness portion of the membrane. The experimental results show that the water permeability constant decreases from 3.15 ± 0.02 to 2.74 ± 0.10 L·m^−2^·h^−1^·bar^−1^ when the intrinsic thickness of the membrane increases by 9 nm. The defects on the film surface generate when the higher concentration of MPD is matched with the relatively low concentration of TMC. In addition, the role of MPD and TMC in the micro-structure of the PA membrane was discussed, which may provide a new way for the preparation of high permeability and high selectivity composite reverse osmosis membranes.

## 1. Introduction

Thin-film composite reverse osmosis membrane (TFC-RO) is the core technology for wastewater treatment, surface water treatment, brackish treatment, and seawater desalination processes [1,2,3,4,5,6]. Typical TFC-RO membranes consist of three layers: a polyester backing fabric, a porous polysulfone (PSF) interlayer, and an ultrathin polyamide (PA) selective layer [3,7,8,9,10,11]. The PA selective layer, formed by the interfacial polymerization reaction of aqueous amine solution and acyl chloride and characterized by a ridge-and-valley structure in the nano- and microscale, is the determinant of the TFC-RO membrane’s selectivity and water flux [3]. Currently, there are many studies focusing on customizing the PA selective layer [7,12,13,14,15,16,17,18,19,20].

Nowadays, a more profound understanding of PA’s structures has been achieved by advanced characterization techniques [7,8,17,18,21,22,23,24]. In addition to this ridge-and-valley morphology on its surface, the cross-section of this exhibits hollow voids in the scale of tens of nanometers [8,10,17,22,25,26,27], the existence of which gives rise to the apparent thickness (*δ_app_*) of the PA layer to an order of magnitude higher than its intrinsic thickness (*δ_int_*) [8,10,25]. The cognitions of the PA layer’s physical structures changed from the homogeneity [28,29,30] to the heterogeneity [7,8,17,20,23,31,32] based on those characterizations. Our recent study revealed that the release of nanobubbles from the aqueous phase shapes the PA layer with nanobubble-shaped voids [7]. In addition, the backside is characterized by open pore-like structures [8,15,22,26,33]; nevertheless, few researchers have analyzed specifically the structural characteristics of the backside of the PA membrane and how the number and size of the pores are related to polymerization conditions. It has been speculated that the ridge-and-valley characteristic structure on the membrane surface may be related to its hollow voids and the pore structures on its backside [34]. Moreover, it was speculated that the leaf-like structures of the PA layer are prone to the formation of defects [35]. However, how are these micro- and nanostructures of PA nanofilm (surface roughness, defects, apparent thickness, intrinsic thickness, hollow void size, the distributions of pores on the backside) [7,8,10,25] related to film permselectivity? How these micro- and nanoscale features could be controlled by experimental parameters remains to be addressed.

Many previous studies demonstrate that the enhancement of roughness and leaf-like structures on the PA’s top surface is positively related to the water flux [7,14,15,16,25,36,37,38,39,40,41,42,43,44,45]. Jiang et al. prepared some ultra-thin films to improve the permeating flux of the film [19,20], which was accomplished by spreading out the PA layer from the crumpled to the smooth on the free organic/aqueous interface. The study proves that water permeance increases with the thickness decrease; however, the PA layer’s thickness measured by AFM was referred to as its entire height (apparent thickness) [20]. In addition, this measurement is not able to exclude the impact of the hollow voids within the PA layer, while the hollow voids do not participate in the separation of salt–water molecules, they are included in the measurement of PA’s overall thickness. Therefore, the actual zone of the PA layer that functions as a barrier to separate the salt and water molecules may just cover its intrinsic thickness parts. Hoek et al. believed that the structure of the surface of the PA membrane is only a byproduct during the IP process, and it is the inner rejection layer of the membrane that plays the key role in osmotic separation [13]. The existence of the internal rejection layer may indicate the chemical heterogeneity and non-uniformity of the PA layer [8,11,13,17,19,29,31,46,47,48]. Freger shows a similar view that there exists a dense core layer in the polyamide and the PA layer’s crosslinking degree varies from one part to another [47]. In addition, the connectivity between the hollow voids inside the PA layer and the pore structure on the backside of the membrane and the presence of defects on the film surface has been previously reported [8,23,46].

Monomer concentrations (i.e., diamine and acyl chloride) are the fundamental and intrinsic determining factors of the interfacial polymerization process. Hence, studying the impact of the monomer concentrations on the membrane’s micro- and nanostructures would provide a foundation for further investigation into the relationship between the latter and the membrane’s performance. Jin et al. pointed out that the variation in the intensity of absorbance peak of the PA layer’s chemical bonds means the different thickness film with the increasing of monomer concentration by using Fourier-Transform Infrared Spectroscopy [49]. However, the thickness measured in this way was the PA’s apparent thickness. Xu et al. systematically carried out a series of experiments via changing the concentration of reaction monomers in the IP process to investigate the relationship between the PA layer’s morphology and its separation performance [22]; it points out that for the RO membrane with excellent performance, its PA layer should meet some demands: the top surface contains a leaf-like structure, porous backside, and the cross-section is full of voids. However, they neglected the existence of film defects which can greatly weaken the film performance of salt rejection.

Based on the above discussion, this article aims to investigate the relationship between the micro- and nanoscale structures of the PA nanofilm and the varied monomer concentration. After that, the correlations between the micro- to nanoscale structures of the PA membrane including the surface roughness, nodule-like and leaf-like morphology characteristics, and the pore structure on the backside of the PA membrane and its composite membrane performance are explored and analyzed.

## 2. Materials and Methods

### 2.1. Materials and Methods

Polysulfone (PSF) ultrafiltration membrane was prepared by laboratory pilot line (molecular weight cut-off: 30,000–50,000 Da). Reacting monomer, m-phenylenediamine (MPD, 99.5%, Aladdin, Shanghai, China), and 1,3,5-benzenetricarbonyl trichloride (TMC, 99%, Aladdin, Shanghai, China) was used to prepare the polyamide rejection layer. (+)-10-camphor sulfonic acid (CSA, 99%, Aladdin, Shanghai, China) was added into the aqueous phase to keep the supporting layer of PSF at a good wettability to avoid the pores whose surface shrunk in the process of membrane drying in the oven, and trimethylamine (TEA, AR, 99%, Aladdin, Shanghai, China) was also added into aqueous phase and functioned as an acid acceptor which reacted with byproduct HCl, then positively promoted the interfacial polymerization. Isopar^®^ G (ExxonMobil Chemical, Houston, TX, USA) was used as an organic solution to dissolve TMC. N, N-Dimethylformamide (DMF, AR, 99.5%, Aladdin, Shanghai, China) was used to dissolve the interlayer PSF to strip the PA layer. Sodium chloride (NaCl, 99.5%, Aladdin, Shanghai, China) was used to test membrane separation selectivity. Sodium hydroxide (NaOH, 96%, Aladdin, Shanghai, China) was used to adjust the pH of the feed salt solution in the process of membrane performance characterization.

### 2.2. Membrane Preparation

The compositions of aqueous and organic solutions for the TFC membranes are summarized in Table 1. We fabricated 3 series of TFC membranes: (1) series 1: The *c*(MPD) varied while the *c*(TMC) was fixed constantly at 0.11%; (2) series 2: The *c*(TMC) varied while the *c*(MPD) was fixed constantly at 2.2%; (3) series 3: Both *c*(TMC) and *c*(MPD) varied at the same time while the ratio of *c*(MPD) to *c*(TMC) was fixed at the value of 20. All three series of membranes follow the same preparation protocol, which is given in Appendix A. Only the MPD and TMC concentrations were varied.

### 2.3. Characterization

#### 2.3.1. Morphological Characterization

The morphology and chemical analysis of the membranes were studied using SEM, TEM, AFM, ATR-FTIR, and XPS. The details are presented in “Characterization methods” in Appendix A.

#### 2.3.2. Evaluation of Membrane Performance

Both the low- and high-salt water were tested by a crossflow reverse osmosis machine. The low salinity conditions contained the following parts: the feed solution was 2 g/L NaCl, the test pressure was 1.55 MPa, the crossflow speed was kept a constant value at 3 L/min, the temperature of feed solution was controlled at 25 ± 1 °C; in addition, 0.1 mol/L sodium hydroxide was added into the feed solution to adjust pH = 7.5 ± 0.5. The high salinity which was simulated to be seawater contained 32 g/L NaCl and 5 mg/L H_3_BO_3_, the test pressure was 5.5 MPa, and the other test conditions were the same as those of low salinity. In addition, the pure water flux was tested under the same conditions as those under low salinity 2 g/L NaCl to obtain the permeability coefficient A (L·m^−2^·h^−1^·bar^−1^). Before the characterization of membrane’s water flux and salt rejection, all of the membranes were stabilized under the corresponding test pressure (1.55 MPa and 5.5 MPa for low- and high-salt water, respectively) for 60 min. When the actual test had begun for 30 min, the permeating solution flowing through each membrane was collected. It was necessary that a conductivity meter should be used to test the permeations to clearly know the concentration of salt solution, and the measuring cylinder was used to read how high the permeation was. The NaCl rejection and permeation flux could be calculated based on the following equations:(1)$$J=\frac{\Delta V}{S\cdot \Delta t}$$
where Δ*V* (L) is the volume of permeation, *S* (m^2^) is defined as the permeating area that water flowed through, Δ*t* is test time, *J* (L·m^−2^·h^−1^) is permeating flux.
(2)$$R=\frac{C_{f}-C_{p}}{C_{f}}\times 100\%$$
where *C_f_* (mg/L) and *C_p_* (mg/L) are the solute NaCl concentration of feed solution and permeation, respectively.
(3)$$A=\frac{J_{w}}{\Delta P}$$
where *J_w_* (L·m^−2^·h^−1^) is the pure water flux; Δ*P* (bar) is the test pressure; *A* (L·m^−2^·h^−1^·bar^−1^) is the permeability coefficient of the water molecule while going through the polyamide membrane.
(4)$$B_{s}=\frac{J_{s}}{C_{f}-C_{p}}=J_{sw}\times \frac{1-R}{R}$$
where *J_sw_* (L·m^−2^·h^−1^) is the water flux including the solvent DI-water and the solute NaCl; *J_s_* (L·m^−2^·h^−1^) refers to solute NaCl flux; *C_f_* and *C_p_* are clearly specified in Equation (2); *B_s_* (L·m^−2^·h^−1^) is the permeability coefficient of the solute NaCl.

Additionally, the calculation of Boron rejection (*R_boron_*) is similar to that of salt NaCl and the boron concentration of the feed and permeating solution are tested by using inductively coupled plasma-optical emission spectrometer (ICP-OES, PerkinElmer, AvioTM 200, Shanghai, China).
(5)$$R_{boron}=\frac{C_{f}-C_{p}}{C_{f}}\times 100\%$$
where the *C_f_* (mg/L) and *C_p_* (mg/L) is the boron concentration of feed solution and permeation, respectively.

## 3. Results and Discussion

### 3.1. The Structural Analysis of the PA Layer

#### 3.1.1. Series 1

Figure 1 shows the top surface of the TFC membranes in series 1. When *c*(MPD) was relatively low (i.e., 0.25~2.2%, TFC−1~4), both of the nodule-like and leaf-like structures were abundant on the PA layer’s top surface. Then, as *c*(MPD) increased, the leaf-like structures gradually disappeared, and the PA top layer was characterized mainly by nodular structures. Meanwhile, it should be noted that as the *c*(MPD) increased, the nodules became larger. As shown in Table 2, the nodular size (*d_n_*) follows the order: TFC−6 (87.90 ± 21.19 nm) > TFC−4 (68.34 ± 12.50 nm) > TFC−1 (25.54 ± 5.74 nm). In some circumstances, the nodules could not sustain their original nanobubble shape and collapsed into “donut-like” and “leaf-like” structures (i.e., collapsed nodules) during the dehydration process after IP process [7,8]. For these structures, the collapsed nodules’ size (*d_c-n_*) (Table 2) was calculated by line analysis on FESEM images. The lateral size of the collapsed structures shows the same variation trend with the *d_n_*. This implies the nodules and the collapsed nodules have the same origin at the initial status.

Theoretically, *δ_c-n_* is approximately twice *δ_int_*. We measured the thickness of the edge (*δ_c-n_*) of the “donut-like” or the leaf-like structures (i.e., collapsed nodules). As Table 2 shows, the *δ_c-n_* is only 20.15 ± 2.89 nm of TFC−1; it gradually increased by 50% to be 29.20 ± 1.91 nm when *c*(MPD) was increased to 1.1%, then further continued to go up to be 45.34 ± 9.31 nm of TFC−6. This observation agrees well with our previous study that these nodules were shaped by confined gas nanobubbles at a nascent state [8]. In addition, occasional pores can be observed on the surface of TFC−6 (Appendix A). The continuous increasing in the *d_n_* and the generated defects at extremely high *c*(MPD) conditions might be related to the faster growth of confined nanobubbles as IP reaction is accelerated by the increase in *c*(MPD) [7,8,15,16]. From both the TEM image and the collapsed “donut-like” nodule [8,25], we conclude that the monomer concentration plays a crucial and significant role in the intrinsic thickness of the PA layer. In addition, we further notice that the “donut-like” or “leaf-like” structures disappeared gradually and finally vanished as the *c*(MPD) was increased. This phenomenon may be related to the higher mechanical strength of the nodules fabricated at high *c*(MPD) (and therefore higher intrinsic thickness), i.e., the thicker wall thickness of the nanobubble at high MPD/TMC concentration prevents the nodules-like structures from collapsing into the “donut-like” structures. This explains why “donut-like” structures were seldom observed in TFC−4 to TFC−6 (Figure 1) and further proves the “nanobubble” theory [7,8,15].

Figure 2 shows the PA back surface corresponding to series 1 in Figure 1. As agreed with our previous study [7,8,25,26], the back surface of the PA nanofilm is porous. Interestingly, the pore size (*d_p_*) and surface porosity (*ϕ_o_*) generally increased as the *c*(MPD) was increased. This indicates that the *d_p_* and *ϕ_o_* values are positively correlated with *c*(MPD) when *c*(TMC) is fixed. This phenomenon can be explained by the fiercer interfacial reaction at high concentrations of MPD and TMC [8]. Moreover, the higher *c*(MPD) is, the more alkaline the MPD solution is [15,16]. The alkalinity favors more CO_2_ dissolving in the MPD solution, therefore facilitating faster releasing of nanobubbles [8], which in turn shape the larger pores on the backside of the PA layer [7,8].

Figure 3 showed the TEM cross-sectional micro-images of TFC−1 to TFC−6. Based on these micro-images, *δ_int_* and *δ_app_* are derived (deriving protocols shown in SI). For lower *c*(MPD) TFC−1 and TFC−2, the TEM micro-images had a much lower contrast compared to TFC−6 with high *c*(MPD). Such a phenomenon might suggest a lower PA layer density (e.g., cross-linking degree) at low *c*(MPD) (and hence electron penetration will be higher). All in all, that experimental phenomenon demonstrates that the lower *c*(MPD)-reacting monomer produces the thinner *δ_int_*. While varying *c*(MPD) from 0.25% to 8.8%, the *δ_int_* increased significantly from 5.65 ± 0.76 nm to 29.75 ± 1.96 nm. In comparison, the *δ_app_* increased from 36.71 ± 14.27 (TFC−1) to 118.29 ± 26.12 (TFC−4), and then gradually decreased to 81.08 ± 16.91 (TFC−6). The *δ_int_* monotonically increases with the increment of *c*(MPD), which elevates the IP reaction [11]. However, the *δ_app_* increases first due to the growth of the nanobubble structures until a further increase in *c*(MPD) restricts the growth of nanobubbles by producing a thicker nanobubble wall thickness [7,8,11]. Hence, the *c*(MPD) plays a vital role in determining the physical structure of the PA layer nanostructures.

#### 3.1.2. Series 2

As shown in Figure 4, when *c*(TMC) was extremely low, the top surfaces of the PA layer showed mostly large collapsed structures, while nodular structures could seldom be observed. For the membranes fabricated with extremely low *c*(TMC) (i.e., TFC−7) and *c*(MPD) (i.e., TFC−1), the edge thickness of the collapsed structures (*δ_c-n_*) on both membranes are thin, which suggests the low concentration of *c*(TMC) or *c*(MPD) limits the growth of the PA nanofilm. However, the lateral size (*d_c-n_*) of the collapsed structures on TFC−7 were extensively larger than that on the TFC−1 membrane (i.e., “donut-like” structures). Moreover, defects could be observed on the surface of TFC−7 (Figure 4a) frequently and even on that of TFC−8 membranes (Figure 4b), which might result from the incomplete interfacial reaction at extremely low *c*(TMC) conditions. In fact, an integral PA nanofilm of TFC−7 could not be obtained after dissolution of the PSF substrate layer, confirming the defective and low crosslinking-degree nature of the TFC−7. As the *c*(TMC) was increased above 0.05%, these defects gradually vanished from the surface, and the large-size “leaf-like” structures were gradually replaced by the “donut-like” structures. Interestingly, as *c*(TMC) was increased to 0.11%, the “leaf-like” structures vanished and the membrane surface was mainly characterized by nodules (Figure 4d). As the *c*(TMC) continued to increase to 0.22% (Figure 4e), the superficial nodules were smaller than TFC−9 and TFC−10 and the “leaf-like” structures could be observed more frequently. Then, when *c*(TMC) was increased to 0.44%, nodules and “leaf-like” structures almost disappeared from the surface and “donut-like” structures developed again and covered most of the membrane surface (Figure 4f).

Figure 5 showed SEM images of the PA’s back surface in series 2. When the *c*(TMC) was 0.02%, the resultant TFC−8 had the largest *d_p_* of 70.08 ± 22.53 nm and *ϕ_o_* of 15.65% among all PA membranes (except for TFC−7, whose PA layer could not be separated in this study due to its fragile characteristics) in series 2. Then, as the *c*(TMC) gradually increased to 0.44%, both the *d_p_* and *ϕ_o_* showed a continuously decreasing trend to 13.16 ± 4.09 nm and 0.47%, respectively. The larger *d_p_* results in more favorable water transport at the PA/PSF interface [7,8], this explains well that decreasing the *c*(TMC) usually results in significantly improved water flux [7,8,22,24,50].

Combining the surface morphological analysis of series 1 and series 2, we could conclude that a higher *c*(MPD)/*c*(TMC) ratio is beneficial for the formation of larger “leaf-like” structures, and these large “leaf-like” structures correspond to a more porous PA backside, while a lower *c*(MPD)/*c*(TMC) ratio favors the formation of smaller nodules and “donut-like” structures, and these structures correspond to a less porous PA backside. It appears deducible that increasing either *c*(MPD) or *c*(TMC) promotes the IP reaction and hence increases the *δ_c-n_* of the PA nanofilm, while increasing the ratio of *c*(MPD)/*c*(TMC) increases the size of nanoscale structures (i.e., nodules (*d_n_*) and collapsed nodules (*d_c-n_*)). Moreover, the *d_p_* value appears to vary correspondingly with the *d_n_* value. For example, both *d_p_* and *d_n_* are larger when *c*(MPD)/*c*(TMC) is increased (i.e., *c*(MPD) was larger or *c*(TMC) was smaller). This mechanism is in agreement with the perception that the role of TMC is as a reaction “inhibitor” of the organic phase [7,10,11,25,31,38,47,51,52]. While the inhibitor concentration was decreased, the activator solution penetrated more into the organic phase, hence resulting in larger *d_p_* and *d_n_*.

Figure 6 showed the TEM cross-sectional micro-images of TFC−7 to TFC−12. From the TEM cross-sectional image, we could observe that the superficial small nodules and “donut-like” structures at higher *c*(TMC) rooted in the initial nodular layer, forming a multi-layered void structure. According to our previous analysis [8], the voids of these exterior nanoscale structures are interconnected with the initial nodular layer, forming an “initial + exterior layer” structure. The development trend in the nanostructures in series 2 indicates that the increase in *c*(TMC) above the value of 0.22% promoted the formation of the exterior layer. The reason could be the combing effect of: (a) the increase in *c*(TMC) leads to more gas release; and (b) the role of TMC as an inhibitor confines the size of the nanobubbles, and as a result, the number of nanobubbles surged, pushing the reaction front further away from the initial water–oil interface and forming a multi-layer of nanovoids inside the PA nanofilm. The formed exterior layers increased the *δ_app_* of the PA layers. Similar to series 1, the increase in *c*(TMC) also increased the *δ_int_*, which increased from 12.19 ± 2.84 nm to 26.71 ± 4.99 nm with increasing *c*(TMC). The TEM cross-sectional image further explains the dependence of *d_n_* and *d_p_* on monomer concentration: while increasing *c*(MPD) prompted the formation of a well-defined nodular layer and then led to larger *d_n_* and *d_p_*, increasing *c*(TMC) gave rise to the continual growth of the PA nanofilm vertically [35,53,54] and the formation of a multi-layered (e.g., nodular layer + exterior layer) structure [8], which in turn reduced the *d_n_* and *d_p_*.

#### 3.1.3. Series 3

Figure 7 showed the PA membranes’ surface of TFC−13 to TFC−18 in series 3, in which the ratio between *c*(MPD) and *c*(TMC) was kept at 20. Obviously, when the monomer concentration in both phases was extremely low, the nodules or “leaf-like” structures on the top surface of the PA layer appeared thin and fragile (the bulk PA nanofilm was extremely prone to breaking when being separated from the PSF substrate). Many visible nanopores could be observed to be randomly distributed on the PA surface (Figure 7a). These defects in the nascent nodular layer could be attributed to the insufficient monomer concentration. However, as they were in a middle concentration level, such as the TFC−15~16, the nodules’ size appeared to be larger and firmer (as shown in Appendix A), *δ_c-n_* increased from 21.60 ± 3.26 nm to 32.51 ± 8.78 nm. As the TMC concentration continued to increase to be above 0.11% (i.e., TFC−17 and TFC−18), the superficial collapsed nodules were gradually replaced by the nodular structures. Such situations were similar to TFC−10~11 in series 2, whose exterior nodules form the exterior PA layer and promote the “vertical growth” of the polyamide layer in the constant supplements of *c*(MPD) and *c*(TMC). Similar to series 2, the exterior layer started to grow substantially when *c*(TMC) was larger than 0.11%. For series 1 with constant *c*(TMC) = 0.11%, even if the *c*(MPD) was increased to 8.8%, the exterior layer did not develop at all (TFC−6). This phenomenon suggests that higher *c*(TMC) mainly favors the formation of the multi-layered structure of the PA nanofilm. This is in agreement with the perception that the PA nanofilm forms in the organic phase, and hence is more sensitive to the variation in *c*(TMC) rather than *c*(MPD).

Figure 8 shows the SEM images of each PA’s backside in series 3, except for TFC−13 because it could hardly be transferred to the silicon wafer. There are many evident visual pores in each SEM image and the number density of pores are at approximately the same level (495 counts/μm^2^ for TFC−14, 502 counts/μm^2^ for TFC−16, 516 counts/μm^2^ for TFC−17, and 472 counts/μm^2^ for TFC−18). Such a morphology characteristic may result from the consistent *c*(MPD)/*c*(TMC) ratio of 20 [19,20,41]. Despite the similar level of number density, the pore size *d_p_* on the back surface of each PA membrane varied. When *c*(MPD) was 0.5%, the *d_p_* was 14.36 ± 4.21 nm on average. With increasing monomer concentrations, the *d_p_* first increased to 35.37 ± 4.43 nm for TFC−16, then dropped to 24.46 ± 10.51 nm for TFC−18, and the percentage of the opening area (*ϕ_o_*) shows a similar trend.

Figure 9 shows the cross-section parts of the membrane prepared in series 3. Combined with Table 2, it is obvious that the *δ_int_* roughly increases with monomer concentrations, a similar trend to that found in series 1 and series 2. Interestingly, when comparing the PA membrane prepared by 8.8% *c*(MPD), the *δ_int_* of TFC−18 is lower than TFC−6. This suggests that the higher *c*(TMC) mainly promoted the formation of multiple layers of nano-voids, increased the *δ_app_*, and therefore benefitted the “vertical growth” of the PA nanofilm. In the meantime, when comparing the PA membrane prepared by 0.44% *c*(TMC), the *δ_int_* of TFC−18 is lower than TFC−12. In this case, it seems that the higher *c*(MPD) also promoted the formation of multiple layers of nano-voids. However, considering that the multiple-layered structure did not appear at all in series 1 (in abundant *c*(MPD) but moderate *c*(TMC) conditions), we could conclude that although *c*(MPD) and *c*(TMC) mutually contribute to the multiple layers of nano-voids because of enhanced gas nanobubbles, high *c*(TMC) is the pre-requisite condition for the formation of multi-layered nano-void structures. Meanwhile, when the multi-layered nano-void structures are formed (enhanced *δ_app_*) at extremely high-*c*(TMC) conditions, the *δ_int_* tends to be reduced. Because when the IP reaction happens further away from the initial water–oil interface, the exterior PA nanofilm grows thinner in contrast to the thicker PA layer (initial layer).

### 3.2. Chemical Structure Characterization

#### 3.2.1. FTIR

FTIR is used to assist in revealing and illustrating the law and chemical structure difference behind those membrane morphologies (Appendix A). In series 1, with the c(MPD) increase, both the peak intensity of amide I (1663 cm^−1^) and amide II (1541 cm^−1^) show a strengthening trend, which probably means more newly formed CO-NH groups. Moreover, the difference of the peak intensity concerning amide II seems to become more evident; this may be attributed to the higher *c*(MPD), which brings out more -NH_2_ on the other hand. However, when *c*(MPD) increases to above 4.4%, the peak intensity begins to change a little, which may indicate the reaction between those two monomers slows down or arrives at the end. In series 2, when the *c*(TMC) increases from 0.01% to 0.11%, the peak intensity of amide I becomes stronger. However, as it keeps increasing to 0.44%, the absorption peak intensity of C=O does not continue to change anymore, indicating the CO-NH of the polyamide have developed well, which agrees with Freger’s standpoint that the excess *c*(TMC), such as that surpassing 0.11%, will restrain the growth of polyamide, called self-limitation [11]. On the other hand, when the *c*(TMC) increases from 0.01% to 0.44%, it is hard to understand why the peak intensity of the amide II at 1541 cm^−1^ keeps getting stronger. In series 3, when the ratio of *c*(MPD) to *c*(TMC) was fixed at 20, with increasing monomer concentrations, both the peak intensity of amide I and amide II increase, implying the CO-NH of the polyamide keeps forming, which may demonstrate that the two-phase monomer concentration ratio of about 20 is appropriate for them to form a suitable polyamide layer.

#### 3.2.2. XPS

Appendix A reveals the elements contained within the PA membrane. The O/N ratio within the superficial depth of 10 nm on the PA surface was analyzed by XPS [55]. Generally, a lower value of O/N indicates a higher crosslinking degree of the PA layer at adjoining regions [19,55,56].

As shown in Appendix A, with increasing *c*(MPD) in series 1, the O/N value of the PA’s topside gradually decreases from 1.26 ± 0.06 to 1.02 ± 0.07. This is because a sufficient amount of *c*(MPD) reacts with residual *c*(TMC), resulting in a more compact PA nanofilm [8]. In series 2, with increasing *c*(MPD), the O/N value increased from 1.03 ± 0.03 to 1.69 ± 0.07. This is partially because higher *c*(TMC) requires the increase in *c*(MPD) accordingly to maintain the reaction ratio. The lack of *c*(MPD) leads to an excess amount of acyl groups. On the other hand, higher *c*(TMC) promotes the formation of multi-layered voids in the PA nanofilm of TFC−12 (Figure 6d). By contrast to the single-layered void structure in the TFC−8 membrane, the multi-layered voids increased the *δ_app_* of the PA nanofilm. Therefore, the topmost selective barrier is further from the aqueous/organic interface. The comparative lower *c*(MPD) and higher *c*(TMC) also contribute to the higher O/N value. This trend is agreeable with the previous studies [8,21,47,57]. In series 3, with the increase in both *c*(MPD) and *c*(TMC), the O/N value shows the same variation trends in series 2. The TFC−16 prepared by 8.8% MPD and 0.44% TMC is quite different from the other membrane, whose cross-section parts contain many hollow voids, and from Table 2, the membrane apparent thickness is the largest one among all 16 polyamide membranes, which also means the existence of second grades or nth grades exterior to the PA layer; the exterior layer will bring out the low cross-linking degree of the PA membrane as discussed in series 2 above.

### 3.3. Evaluation of Membrane Performance

The detailed experimental results are clearly shown in Appendix A. In series 1, as Figure 10a shows, the membrane’s water permeability (*A*) increases from 1.31 ± 0.18 L·m^−2^·h^−1^·bar^−1^ in TFC−1 to 3.15 ± 0.02 L·m^−2^·h^−1^·bar^−1^ in TFC−4, then sharply drops down to 2.74 ± 0.10 L·m^−2^·h^−1^·bar^−1^ in TFC−6. However, the *R_a_* of TFC−6’s top surface is bigger than that of TFC−4 and TFC−5 (Table 2), which is in contradiction with previous reports that the enhancement of surface roughness could significantly bring out the promotions of water flux [7,8,36,37,38,40,44,58]. Herein, the difference is that the intrinsic thickness (*δ_int_*) of the PA layer changed greatly from TFC−4 to TFC−6; the *δ_int_* climbs from 21.00 ± 2.54 nm to 29.75 ± 1.96 nm, an increase of approximately 9 nm. This could account for the decrease in *A* value, which also implies that the actual hydraulic resistance to water and salt molecule transport across the PA layer comes from the membrane intrinsic parts (just about 10~30 nm according to this paper). As *c*(MPD) increases from 0.25% to 2.2%, both the *A* value and *δ_int_* show a continuous climbing tendency; nevertheless, based on Table 2, the *R_a_* of PA’s top surface also increases significantly, which can magnify the actual area that water contacts the membrane surface. Moreover, on the PA’s backside, the pore size of TFC−4 is far bigger than that of TFC−1~3, which reduces the intrinsic parts the molecules need to transport from PA’s frontside to its backside. It is the contributions of the big *R_a_* and large *d_p_* that could get over the resisting effects of increased intrinsic thickness to boost water flux. Therefore, the analysis of the membrane’s water permeability should be judged by full-scale aspects: *δ_int_*, *R_a_*, and *d_p_*. Additionally, we obtained the correlation coefficients (*R^2^*) between the membrane’s structure and performance based on series 1 shown in Table 3. The strong correlations between *c*(MPD) and *δ_int_*, *R_a_*, *d_p_* could be found, which are 0.87, 0.99, 0.97, respectively; these agree well with our preceding reports on the backside’s pore size, and the ridge-and-valley morphologies on its top surface are violently sharpened by confined nanobubbles sandwiched between the PA layer and the supporting layer [7]. The fast releasing of nanobubbles induced by a more alkaline aqueous solution directly aggravates that sharpening effect, with the increase in *c*(MPD) [8,16,40]. On the other hand, for the membrane’s selectivity, it is amazing that the *B_s_* suddenly begins to climb to a high value of 9.95 ± 0.40 L·m^−2^·h^−1^ and the *R_boron_* drops to only 61.87 ± 1.15%; the integrity of TFC−6’s top surface, whose O/N is as low as 1.02 ± 0.07, would not be responsible for the great decrease in the membrane’s selectivity, which could be explained by the generated defects, such as those shown in the image in Figure 10c. This phenomenon supports our previous study on the defects on the membrane’s surface occurring when PA’s intrinsic parts could not package the fast release of nanobubbles [7,8]. Moreover, some defects located in other places are captured in Appendix A.

In series 2, the water permeability (*A*) of the membrane continues to decrease, varying from 7.64 ± 1.27 L·m^−2^·h^−1^·bar^−1^ to 1.76 ± 0.01 L·m^−2^·h^−1^·bar^−1^. The water permeability of TFC−7 is close to that of the supporting layer PSF, which means the formation of an imperfect polyamide film. The *δ_int_* of the film increases from 12.19 ± 2.46 nm to 26.71 ± 4.99 nm, and there is a strong negative correlation (*R^2^* = −0.96) between water permeability and *δ_int_*, as shown in Table 4. In addition, those results support the previous research that slightly reducing organic phase *c*(TMC) can increase the water flux [48,50]. With increasing *c*(TMC), the TMC monomer does not only participate in the interfacial polymerization to react with the MPD monomer, but also inhibits it to further diffuse into the organic phase and a rougher surface can hardly be produced [11,47]; on the other hand, the reaction moves towards the water–oil phase interface and the *d_p_* shrinks to be small, as shown in Table 2. The fixation of the amine concentration directly limits the release rate of the nanobubbles. Even when a large amount of TMC is added, the surface of the membrane only tends to generate small nodular structures with less roughness. Regarding membrane selectivity, TFC−7 and TFC−8 have far a greater *A* value than ordinary RO membranes, but their selectivity is extremely low: the *B_s_* is as high as 35.21 ± 21.75 L·m^−2^·h^−1^ and the *R_boron_* is only 57.49 ± 12.79% correspondingly, which may be attributed to the incompleteness of the polyamide film caused by the surface defects [7,8] of the film, as labeled in Figure 11b. Moreover, the defects on the membrane’s topside will be more evident in some extreme environments, as shown in Appendix A. In addition, when the *c*(TMC) increases from 0.22% to 0.44%, the *B_s_* increases from 0.15 ± 0.01% to 0.49 ± 0.08%, and *R_boron_* drops from 76.01 ± 0.71% to 60.59 ± 4.22%, which may be attributed to the second layer within the PA layer which is far from the water–oil interface; its formation process has been discussed in detail in the previous discussion. The crosslinking degree of this second layer is much lower than its initial layer; these results can explain the sharp decline in membrane selectivity.

In series 3, the inference that an increase in the membrane’s intrinsic thickness significantly reduces the water permeability could also be confirmed. Described in Figure 12, from TFC−13 to TFC−18, the *A* value decreases from 17.45 ± 2.07 L·m^−2^·h^−1^·bar^−1^ to 0.79 ± 0.12 L·m^−2^·h^−1^·bar^−1^, while the membrane’s *δ_int_* increases from 6.66 ± 1.95 nm to 23.41 ± 3.52 nm; the closely negative correlation between the *A* value and the *δ_int_* is −0.91, as shown in Table 5. Additionally, especially for TFC−16 prepared by high *c*(MPD) and *c*(TMC) where the secondary structures (seen in Figure 6) have grown up even if they could increase the membrane’s *R_a_*, the *A* value still drops evidently (the relative coefficients between the *A* value and the *R_a_* and the *d_p_* are -0.45 and -0.48, showing weak and poor correlations as Table 5 shows). However, the *B_s_* continues to decline from 17.89 ± 9.06 L·m^−2^·h^−1^ to 0.15 ± 0.09 L·m^−2^·h·^−1^, and the *R_boron_* increases rapidly from 0.92 ± 0.91% to 75.84 ± 0.59%. Nanoscale pores are no longer present on the surface of the membrane except for TFC−13 (seen in Figure 7), which means that membrane integrity is continuously improving. For the membrane’s salt permeability, from TFC−17 to TFC−18, the NaCl rejection (*R_NaCl_*) falls (seen in Appendix A), which may result from the secondary structure, whose low crosslinking degree increases the *B_s_*. However, the *R_boron_* still increases; this may be due to such close similarity between boric acid and water, including their molecular structure, molecular size, etc. While the water molecules could transport through the PA layer, the boric acid could do the same.

### 3.4. Correlation Analysis of the Membrane Structure and Properties

Table 3 shows the correlation between each parameter of the TFC membrane corresponding to different concentrations of MPD when *c*(TMC) is fixed. As can be seen from the table, there is a strong correlation between *c*(MPD) and *B_s_*, indicating that a high concentration of MPD is beneficial to improving the retention performance of the membranes. *c*(MPD) and *δ_int_*, *R_a_*, and *d_p_* have correlations of 0.87, 0.99, 0.97, indicating that the highly alkaline aqueous phase conditions of a high concentration of amine promote the rapid release of nanobubbles, and also intensify the modification of the structure and surface of the polyamide membranes. The correlation between the *A* values and *R_a_* was only 0.55, which did not show a strong correlation, which shows that when the intrinsic thickness of the membrane is changed, *A* values should comprehensively consider the intrinsic thickness, surface roughness, and the pore size of the backside. In addition, the correlation between *R_a_* and *d_p_* is 0.95, which indicates a good correspondence between the structure of the membrane surface and the pore structure of the membrane backside.

Table 4 shows that when *c*(MPD) is fixed, the *A* value continues to decrease as *c*(TMC) increases, while the intrinsic thickness of the membrane continues to increase. The correlation coefficient of −0.96 shows a strong negative correlation, which reinforces that the permeation resistance of water molecules originates from the portion of homogeneous intrinsic thickness, and increasing the intrinsic thickness of the membrane will increase the permeation resistance of the membrane. However, the apparent thickness of the membrane increases first and then decreases, which corresponds neither linearly nor approximately linearly to the decrease in membrane permeability. *c*(TMC) shows the correlation coefficients of −0.68 with *R_a_* of the membrane surface and −0.80 with *d_p_* of the backside, suggesting that the organic phase monomer has the effect of shrinking the reaction region, resulting in a PA membrane with relatively small roughness.

Table 5 shows a correlation coefficient of −0.91 between *A* value and *δ_int_* when *c*(MPD):*c*(TMC) = 20:1, indicating a strong negative correlation between the two; the increase in the intrinsic thickness of the PA layer decreases the water permeability. As the concentration of the two-phase monomer grows, *B_s_* decreases and *R_boron_* continues to increase, indicating that the membrane compactness is enhanced. The correlation coefficient with the *A* value is −0.94, which indicates that the increase in concentration has an inhibitory effect on water permeability.

## 4. Conclusions

(1) The hydraulic resistance that molecules transport through the PA layer really comes from its intrinsic thickness parts; (2) *c*(MPD) with *δ_int_*, *R_a_*, and *d_p_* have strong correlations of 0.87, 0.99, 0.97, while *c*(TMC) shows correlation coefficients of 0.90, −0.68, and −0.80, respectively, with *δ_int_*, *R_a_*, and *d_p_*. Consequently, high *c*(MPD) with low *c*(TMC) tends to intensify the modification of the structure and surface of the polyamide membranes; (3) Some defects which can extremely lower the salt rejection on the top surface of the PA membrane will generate while there are confined nanobubbles quickly releasing; for example, it happens where the relatively high *c*(MPD) reacts with the relatively low *c*(TMC), compared with the traditional monomer concentration ratio that *c*(MPD)/*c*(TMC) = 20.

## Figures and Tables

**Figure 1 membranes-12-00417-f001:**
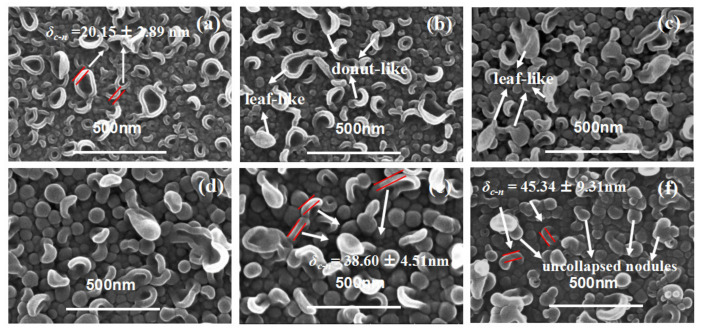
SEM image of PA’s top surface. (**a**–**f**): TFC−1~TFC−6.

**Figure 2 membranes-12-00417-f002:**
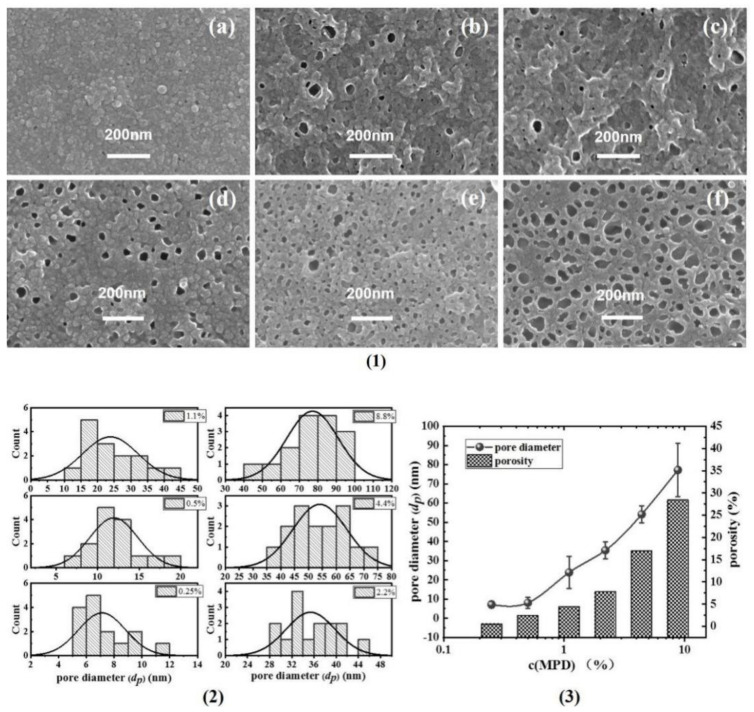
(**1**) SEM images of PA’s back surface, (**a**–**f**): TFC−1~TFC−6; (**2**) the statistics of the pore diameter (*d_p_*); (**3**) the summarizing of the pore diameter and porosity (*φ_o_*).

**Figure 3 membranes-12-00417-f003:**
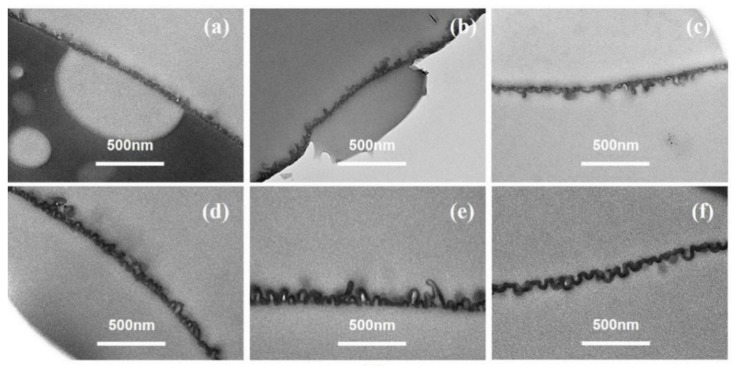
TEM image of PA’s cross-section in series1. (**a**–**f**): TFC−1~TFC−6.

**Figure 4 membranes-12-00417-f004:**
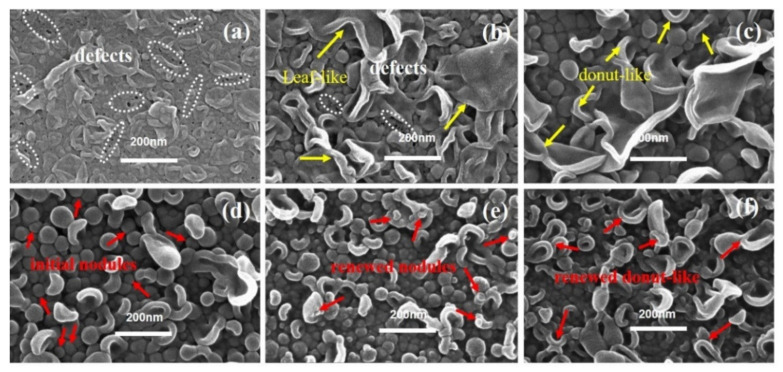
SEM image of PA’s top surface in series 2. (**a**–**f**): TFC−7~TFC−12.

**Figure 5 membranes-12-00417-f005:**
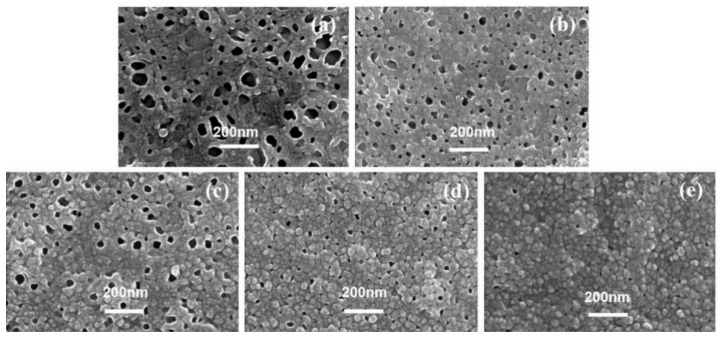
SEM image of PA’s back surface, (**a**–**e**): TFC−8~TFC−12.

**Figure 6 membranes-12-00417-f006:**
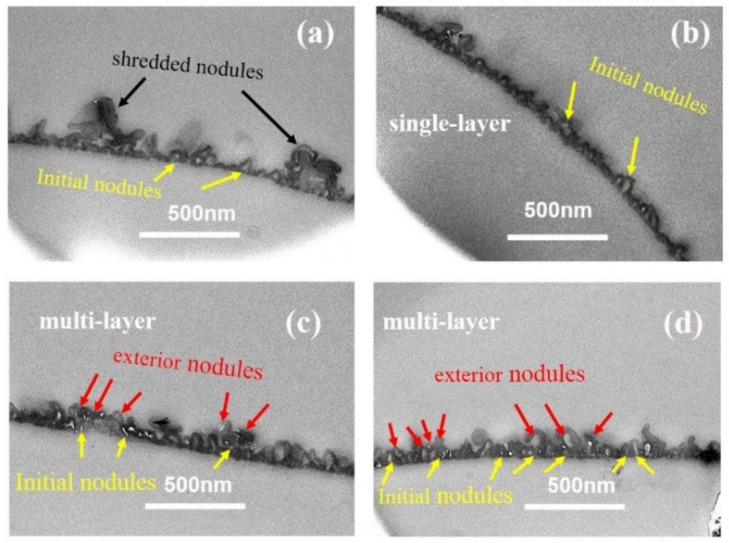
TEM image of cross-section parts of the PA membrane. (**a**–**d**): TFC−9~TFC−12; the cross-section of TFC−7 and TFC−8 could not be achieved.

**Figure 7 membranes-12-00417-f007:**
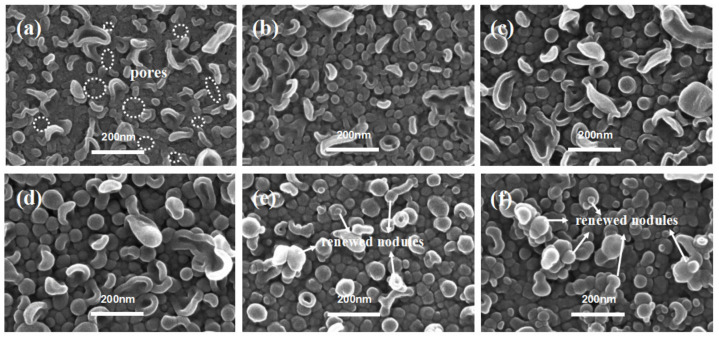
SEM images of PA’s top surface in series 3. (**a**–**f**): TFC−13~TFC−18.

**Figure 8 membranes-12-00417-f008:**
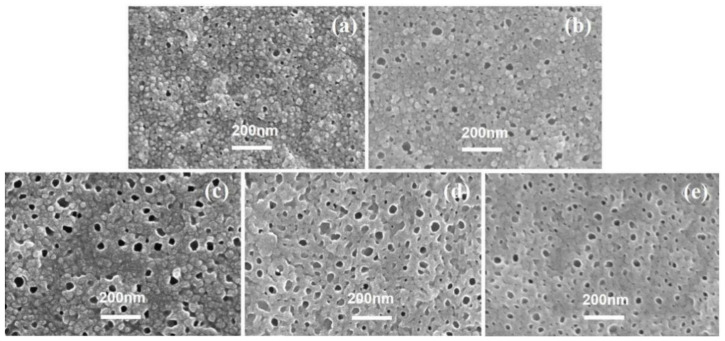
SEM images of PA’s backside in series 3, (**a**–**e**): TFC−14~TFC−18.

**Figure 9 membranes-12-00417-f009:**
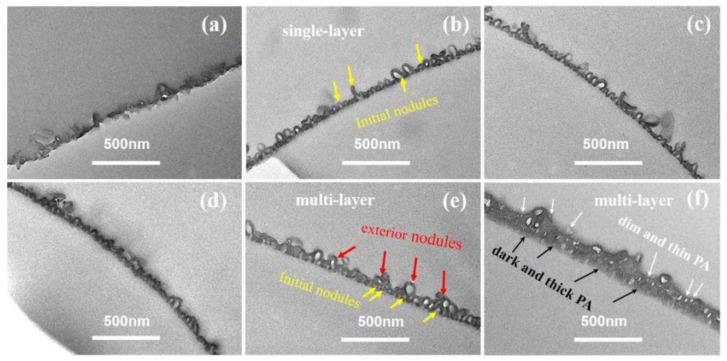
TEM images of cross-section parts of the PA membrane, (**a**–**f**): TFC−13~TFC−18.

**Figure 10 membranes-12-00417-f010:**
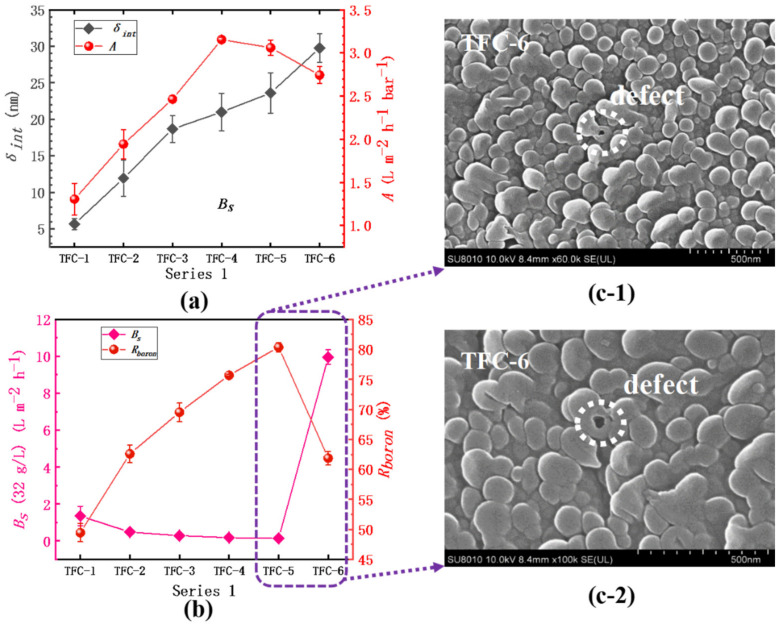
Series 1. (**a**) The variation in water permeability (*A*) and the intrinsic thickness (*δ_int_*); (**b**) the analysis of the salt permeability (*B_s_*) and the boron rejection (*R_boron_*); (**c-1**) and (**c-2**) SEM image of TFC−6.

**Figure 11 membranes-12-00417-f011:**
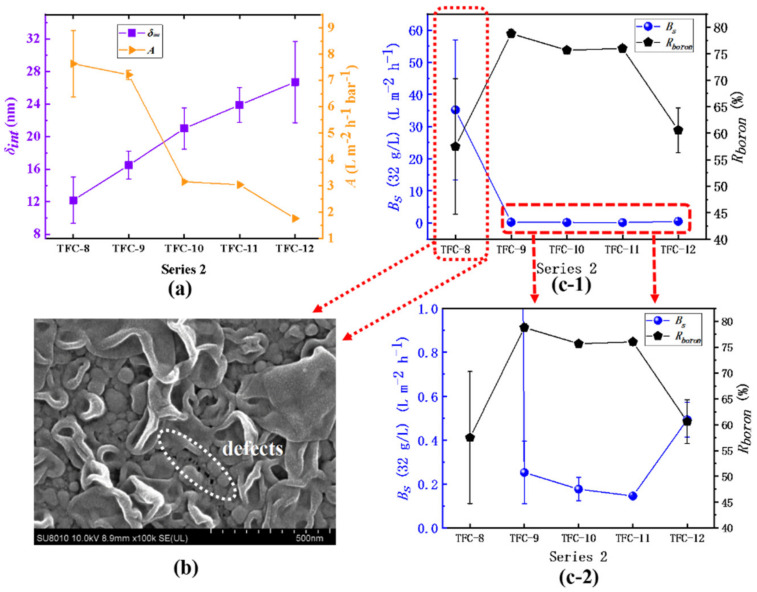
Series 2: (**a**) the variation in the *A* value and the *δ_int_*; (**b**) SEM image of TFC−8; (**c-1**) and (**c-2**) the analysis of the *B_s_* and the *R_boron_*.

**Figure 12 membranes-12-00417-f012:**
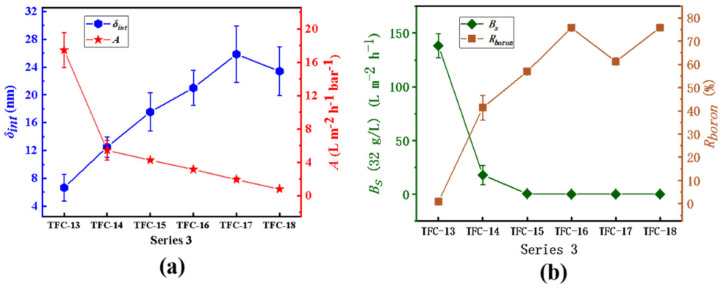
Series 3. (**a**) the variation in *A* value and the *δ_int_*; (**b**) the analysis of the *B_s_* and *R_boron_*.

**Table 1 membranes-12-00417-t001:** Membranes prepared by different reacting monomer concentration.

	Membrane	MPD (wt/v%)	TMC (wt/v%)	*c*(MPD)/*c*(TMC)
Series 1 ^a^	TFC−1	0.25	0.11	2.27
	TFC−2	0.5	0.11	4.55
	TFC−3	1.1	0.11	10.00
	TFC−4	2.2	0.11	20.00
	TFC−5	4.4	0.11	40.00
	TFC−6	8.8	0.11	80.00
Series 2 ^b^	TFC−7	2.2	0.01	220.00
	TFC−8	2.2	0.02	110.00
	TFC−9	2.2	0.05	44.00
	TFC−10 ^d^	2.2	0.11	20.00
	TFC−11	2.2	0.22	10.00
	TFC−12	2.2	0.44	5.00
Series 3 ^c^	TFC−13	0.25	0.0125	20.00
	TFC−14	0.5	0.025	20.00
	TFC−15	1.1	0.055	20.00
	TFC−16 ^d^	2.2	0.11	20.00
	TFC−17	4.4	0.22	20.00
	TFC−18	8.8	0.44	20.00

^a^: Series 1 refers to the series of membranes with varied *c*(MPD) while fixing *c*(TMC) at 0.11%; ^b^: Series 2 refers to the series of membranes with varied *c*(TMC) while fixing *c*(MPD) at 2.2%; ^c^: Series 3 refers to the series of membranes with varied *c*(MPD) and *c*(TMC) while fixing *c*(MPD)/*c*(TMC) at 20; ^d^: The TFC−10 and TFC−16 has identical recipe to the TFC−4 membrane.

**Table 2 membranes-12-00417-t002:** The morphological statistics of the features of PA membrane.

Series	NO.	Topside				Cross-Section		Backside	
		*δ_c-n_ ^a^* (nm)	*R_a_ ^b^* (nm)	*d_n_ ^c^* (nm)	*d_c-n_ ^d^* (nm)	*δ_int_ ^e^* (nm)	*δ_app_ ^f^* (nm)	*d_p_ ^g^* (nm)	*ϕ_o_ ^h^* (%)
**1#**	TFC−1	20.15 ± 2.89	19.78 ± 0.95	25.54 ± 5.74	50.3 ± 9.04	5.65 ± 0.76	36.71 ± 14.27	7.12 ± 1.69	0.51
	TFC−2	26.00 ± 1.69	19.7 ± 0.73	33.30 ± 4.57	57.00± 14.90	11.93 ± 2.46	39.83 ± 9.06	8.00 ± 2.89	2.43
	TFC−3	29.20 ± 1.91	20.30 ± 0.68	50.93 ± 11.66	76.03 ± 14.13	18.67 ± 1.86	41.00 ± 8.45	23.84 ± 8.35	4.41
	TFC−4	35.40 ± 6.24	23.60 ± 1.25	68.34 ± 12.50	87.26 ± 23.87	21.00 ± 2.54	118.29 ± 26.12	35.37 ± 4.43	7.82
	TFC−5	38.60 ± 4.51	24.70 ± 0.72	80.40 ± 12.00	139.22 ± 34.77	23.61 ± 2.77	103.64 ± 41.47	54.11 ± 4.53	16.96
	TFC−6	45.34 ± 9.31	32.52 ± 8.19	87.90 ± 21.19	/	29.75 ± 1.96	81.08 ± 16.91	77.19 ± 13.98	28.40
**2#**	TFC−7	15.41 ± 1.73	16.64 ± 0.88	/	144.68 ± 18.13	/	/	/	/
	TFC−8	18.80 ± 3.41	60.02 ± 3.60	42.71 ± 7.77	201.85 ± 81	12.19 ± 2.84	40.00 ± 15.00	70.08 ± 22.53	15.65
	TFC−9	23.89 ± 2.39	61.62 ± 2.07	68.83 ± 12.50	225.88 ± 79.58	16.48 ± 1.71	78.91 ± 29.26	35.73 ± 5.98	9.13
	TFC−10	35.40 ± 6.24	23.60 ± 1.25	68.34 ± 12.50	87.26 ± 23.87	21.00 ± 2.54	118.29 ± 26.12	35.37 ± 4.43	7.82
	TFC−11	30.39 ± 2.34	30.24 ± 2.21	55.08 ± 11.67	94.2 ± 12.9	23.89 ± 2.12	83.76 ± 24.15	19.33 ± 4.65	1.72
	TFC−12	22.72 ± 2.82	27.04 ± 1.31	29.54 ± 18.36	98.92 ± 26.63	26.71 ± 4.99	97.96 ± 24.12	13.16 ± 4.09	0.47
**3#**	TFC−13	21.60 ± 3.26	21.86 ± 1.12	38.43 ± 6.53	80.03 ± 9.89	6.66 ± 1.95	50.68 ± 11.93	/	/
	TFC−14	25.63 ± 2.13	22.20 ± 0.20	44.23 ± 8.77	77.50 ± 13.15	12.51 ± 1.47	68.73 ± 18.77	14.36 ± 4.21	2.67
	TFC−15	28.66 ± 1.95	42.38 ± 5.82	62.45 ± 11.99	150.77 ± 35.69	17.55 ± 2.77	72.42 ± 22.51	27.33 ± 8.88	5.83
	TFC−16	35.40 ± 6.24	23.60 ± 1.25	68.34 ± 12.50	87.26 ± 23.87	21.00 ± 2.54	118.29 ± 26.12	35.37 ± 4.43	7.82
	TFC−17	42.49 ± 8.04	27.06 ± 1.48	77.92 ± 13.22	/	25.85 ± 4.03	100.07 ± 18.14	31.39 ± 6.52	20.64
	TFC−18	32.51 ± 8.78	45.02 ± 4.83	66.29 ± 9.90	/	23.41 ± 3.52	159.51 ± 23.71	24.46 ± 10.51	16.96

*^a^ δ_c-n_* (nm): the width of the fringe of the collapse nodules measured based on 15 positions of PA top surface in SEM images; *^b^ R_a_* (nm): average roughness of PA top surface observed by its 5 locations in AFM images; *^c^ d_n_* (nm): nodules’ diameter measured by 15 positions of PA’s surface in SEM images; *^d^ d_c-n_* (nm): collapsed nodules (leaf-like or donut-like morphology) measured by 15 positions of PA’s surface in SEM images; *^e^ δ_int_* (nm): the intrinsic thickness obtained by 15 locations of PA cross-section part in TEM images; *^f^ δ_app_* (nm): the apparent thickness obtained by 15 locations of PA cross-section part in TEM images; *^g^ d_p_* (nm): the pore diameter of its back surface measured by 15 locations in SEM images; *^h^ ϕ_o_* (%): the percentage of opening area of its back surface analyzed by image J based on the SEM images. (The detailed measurements are the same as our previous studies [8,25] and we show them in Appendix A).

**Table 3 membranes-12-00417-t003:** The analysis of correlation coefficients in series 1.

1#	*c*(MPD) (%)	*A* (L·m^−2^·h^−1^·bar^−1^)	*B_s_* (L·m^−2^·h^−1^)	*δ_int_* (nm)	*d_p_* (nm)	*R_a_* (nm)	*δ_app_* (nm)
*c*(MPD) (%)	1.00						
*A* (L·m^−2^·h^−1^·bar^−1^)	0.55	1.00					
*B_s_* (L·m^−2^·h^−1^)	0.85	0.10	1.00				
*δ_int_* (nm)	0.87	0.86	0.56	1.00			
*d_p_* (nm)	0.97	0.71	0.71	0.94	1.00		
*R_a_* (nm)	0.99	0.55	0.87	0.85	0.95	1.00	
*δ_app_* (nm)	0.50	0.87	0.08	0.67	0.65	0.54	1.00

*^a^* correlation coefficients *R^2^* were analyzed with the help of Excel. ① |*R^2^*| ≥ 0.8: a strong correlation among two figures; ② 0.8 ≥ |*R^2^*| ≥ 0.5: strong correlations; ③ 0.5 ≥ |*R^2^*| ≥ 0.3: weak correlations; ④ |*R^2^*| ≤ 0.3: no obvious correlation.

**Table 4 membranes-12-00417-t004:** The analysis of correlation coefficients in series 2.

2#	*c*(TMC) (%)	*A* (L·m^−2^·h^−1^·Bar^−1^)	*B_s_* (L·m^−2^·h^−1^)	*δ_int_* (nm)	*R_a_* (nm)	*d_p_* (nm)	*δ_app_* (nm)
*c*(TMC) (%)	1.00						
*A* (L·m^−2^·h^−1^·bar^−1^)	−0.84	1.00					
*B_s_* (L·m^−2^·h^−1^)	−0.48	0.64	1.00				
*δ_int_* (nm)	0.90	−0.96	−0.76	1.00			
*R_a_* (nm)	−0.68	0.96	0.58	−0.86	1.00		
*d_p_* (nm)	−0.80	0.82	0.89	−0.94	0.69	1.00	
*δ_app_* (nm)	0.45	−0.76	−0.85	0.73	−0.80	−0.72	1.00

*^a^* correlation coefficients *R^2^* were analyzed with the help of Microsoft Excel software. ① |*R^2^*| ≥ 0.8: a strong correlation among two figures; ② 0.8 ≥ |*R^2^*| ≥ 0.5: well correlations; ③ 0.5 ≥ |*R^2^*| ≥ 0.3: weak correlations; ④ |*R^2^*| ≤ 0.3: no obvious correlations.

**Table 5 membranes-12-00417-t005:** The analysis of correlation coefficients in series 3.

3#	*c*(MPD) (%)	*c*(TMC) (%)	*A* (L·m^−2^·h^−1^·Bar^−1^)	*B_s_* (L·m^−2^·h^−1^)	*δ_int_* (nm)	*R_a_* (nm)	*d_p_* (nm)	*δ_app_* (nm)
*c*(MPD) (%)	1.00							
*c*(TMC) (%)	1.00	1.00						
*A* (L·m^−2^·h^−1^·bar^−1^)	−0.94	−0.94	1.00					
*B_s_* (L·m^−2^·h^−1^)	−0.49	−0.50	0.72	1.00				
*δ_int_* (nm)	0.72	0.72	−0.91	−0.82	1.00			
*R_a_* (nm)	0.55	0.54	−0.45	−0.50	0.24	1.00		
*d_p_* (nm)	0.15	0.16	−0.48	−0.86	0.72	0.01	1.00	
*δ_app_* (nm)	0.91	0.91	−0.88	−0.54	0.67	0.40	0.33	1.00

*^a^* correlation coefficients *R^2^* were analyzed with the help of Microsoft Excel software. ① |*R^2^*| ≥ 0.8: there exists a strong correlation among two figures; ② 0.8 ≥ |*R^2^*| ≥ 0.5: strong correlations; ③ 0.5 ≥ |*R^2^*| ≥ 0.3: weak correlations; ④ |*R^2^*| ≤ 0.3: no obvious correlations.

## Data Availability

The data presented in this study are openly available at FigShare https://figshare.com/.

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
