# Peer review of "The Intrinsic Parameters of the Polyamide Nanofilm in Thin-Film Composite Reverse Osmosis (TFC-RO) Membranes: The Impact of Monomer Concentration"

_membranes, 2022, doi:10.3390/membranes12040417_

Round 1
Reviewer 1 Report
Dear Authors, The article titled "The intrinsic parameters of the polyamide nanofilm in thin-film composite reverse osmosis (TFC-RO) membranes: the impact of monomer concentration" is nice read but reviewer has following comments. (1) Similar types of work have been done before. The article lacks originality. (2) Reviewer tried to find the originality of the work. (3) The recent review about the RO membrane describes similar type of material and discusses about the parameters affecting the performance of the membranes. Please refer to this article: https://doi.org/10.1016/j.desal.2021.114939
Author Response
We sincerely thank the editor and all reviewers for their valuable feedback that we have used to improve the quality of our manuscript. We have tried our best to revise the manuscript. The reviewer comments are laid out below in italicized font and specific concernshave been numbered. Our response is given in blue normal font, and changes/additions to the manuscript are given in red text.
Response to reviewer 1
Comments and Suggestions for Authors
Dear Authors, The article titled "The intrinsic parameters of the polyamide nanofilm in thin-film composite reverse osmosis (TFC-RO) membranes: the impact of monomer concentration" is nice read but reviewer has following comments.
Response:
We greatly appreciate the reviewer’s circumspection and constructive comments. Please refer to the point-to-point response for detailed explanation and revisions.
Comment 1: Similar types of work have been done before. The article lacks originality.
Response: We appreciate for reviewer’s comment. As for the novelty, it is known that monomer concentrations are the fundamental and intrinsic determining factors of the interfacial polymerization process, which can affect the membrane performance and structure. Although there are some reports focusing on the thickness adjusted by the monomers concentration, the focus on intrinsic thickness has been rarely explored. In this report, we emphasized the effect of intrinsic thickness. Besides, we systematically analyzed the correlation between the concentration of the reaction monomer with parameter of permselectivity and micro- and nano- structure.
Comment 2: Reviewer tried to find the originality of the work.
Response: Thanks for the reviewer’s opinion. The introduction of our previous manuscript did not make the novelty explicit, we have strengthened the introduction to emphasize the novelty. The innovative points of this paper are:
- This study confirms that the true permeation resistance of water molecules originates from the intrinsic thickness portion of the membrane. Deepens the understanding of thickness in current researches.
- The correlation between c(MPD), c(TMC) and other parameters of permselectivity and structure was systematically analyzed.
Comment 3: The recent review about the RO membrane describes similar type of material and discusses about the parameters affecting the performance of the membranes. Please refer to this article: https://doi.org/10.1016/j.desal.2021.114939
Response: Thank you for the recommendation.The emphasis of the review is placed on the support membrane used and the reaction conditions and the use of nonreactive additives in the reaction solution. These variables are compared by investigating their effect on the membrane desalination performance. However, our work focuses on analyzing the effect of concentration on the parameters of membrane permselectivity and micro- and nano- structure, which is very different from the review. Thanks for the reference again, which are now included in the revised manuscript and listed as follows:
- Habib, S.; Weinman, S.T. A review on the synthesis of fully aromatic polyamide reverse osmosis membranes. Desalination 2021, 502, doi:10.1016/j.desal.2021.114939.
Reviewer 2 Report
Comments on the manuscript titled “The intrinsic parameters of the polyamide nanofilm in thin-film composite reverse osmosis (TFC- RO) membranes: the impact of monomer concentration” and written by Mengling Zhang et al. is interesting and it is well written and structured. I recommend a minor revision based on the following comments:
- Table 1 seems to be cut
- In section 2.3.2. The authors should be more specific about the solute concentration used, pH, flux recovery and Equation used to determine the permeability coefficients (A and B)
- Could the authors improve the Figures 9a, 9b, 10a,10c-1,10c-2, 11?
Author Response
We sincerely thank the editor and all reviewers for their valuable feedback that we have used to improve the quality of our manuscript. We have tried our best to revise the manuscript. The reviewer comments are laid out below in italicized font and specific concernshave been numbered. Our response is given in blue normal font, and changes/additions to the manuscript are given in red text.
Response to reviewer 2
Comments on the manuscript titled “The intrinsic parameters of the polyamide nanofilm in thin-film composite reverse osmosis (TFC- RO) membranes: the impact of monomer concentration” and written by Mengling Zhang et al. is interesting and it is well written and structured. I recommend a minor revision based on the following comments:
Response:
We feel great thanks for your professional review work on our article. As you are concerned, there are several problems that need to be addressed. According to your nice suggestions, we have made extensive corrections to our previous manuscript, the detailed corrections are listed below.
Comment 1: Table 1 seems to be cut
Response: Thank you for the reviewer’s opinion. We have contacted with the editor about this mistake, this is just a rough typesetting version, the typesetting details will be strengthened later.
Comment 2: In section 2.3.2. The authors should be more specific about the solute concentration used, pH, flux recovery and Equation used to determine the permeability coefficients (A and B)
Response: Thanks for the reviewer’s comment and question. We previously placed these in the supporting material to shorten the pages. We have put it in our manuscript again in this revision.
Additions: The low salinity conditions contained the following parts: the feed solution was 2 g/L NaCl, the test pressure was 1.55 MPa, the crosflow speed kept a constant value at 3 L/min, the temperature of feed solution was controlled at 25±1 ℃, besides, 0.1 mol/L sodium hydroxide were added into the feed solution to adjust pH = 7.5±0.5. the high salinity which was simulated to seawater contained 32 g/L NaCl and 5 mg/L H3BO3, the test pressure is 5.5 MPa, the other test conditions are the same as that of low salinity. Besides, the pure water flux was tested under the conditions the same as that were under low salinity 2 g/L NaCl to obtain the permeability coefficient A (L·m-2·h-1·bar-1).
The NaCl rejection and permeation flux could be calculated based on the following equations, it was necessary that conductivity meter should be used to test the permeations to clearly know the concentration of salt solution and measuring cylinder was used to read how many the permeation was.
(1)
Where ΔV (L) is the volume of permeation, S (m2) is defined as the permeating area that water flowing through, Δt is test time, J (L·m-2·h-1) is permeating flux.
(2)
Where the Cf (mg/L) and Cp (mg/L) is the solute NaCl concentration of feed solution and permeation respectively.
(3)
Where the Jw (L·m-2·h-1) is the pure water flux; ΔP (bar) is the test pressure; A (L·m-2·h-1·bar-1) is permeability coefficient of the water molecule while going through the polyamide membrane.
(4)
Where the Jsw (L·m-2·h-1) is the water flux including the solvent DI-water and the solute NaCl; The Js (L·m-2·h-1) is referred as solute NaCl flux; Cf and Cp are just clearly specified in equation (2); Bs (L·m-2·h-1) is the permeability coefficient of the solute NaCl.
Besides, the calculation of Born rejection (Rborn) is similar to that of salt NaCl and the born concentration of the feed and permeating solution are tested by using inductively coupled plasma-optical emission spectrometer (ICP-OES, PerkinElmer, AvioTM 200, Shanghai, China).
(5)
Where the Cf (mg/L) and Cp (mg/L) is the born concentration of feed solution and permeation respectively.
Comment 3: Could the authors improve the Figures 9a, 9b, 10a,10c-1,10c-2, 11?
Response: We appreciate the reviewer for presenting this point to us. We have revised the figures in this revision.Besides, we have modified the numeration of these figures.
Round 2
Reviewer 1 Report
Dear Author,
The article titled "The intrinsic parameters of the polyamide nanofilm in thin-film composite reverse osmosis (TFC-RO) membranes: the impact of monomer concentration" is interesting now and the author has incorporated the changes needed as suggested earlier. The article is ready to be accepted